# Changes in the place of death before and during the COVID-19 pandemic in Japan

**Masashi Shibata**[1], **Yuki Otsuka**[2]*, **Hideharu Hagiya**[2], **Toshihiro Koyama**[3],
**Hideyuki Kashiwagi**[4], **Fumio Otsuka**[2]

**1** Department of General Medicine, Iizuka Hospital, Iizuka, Japan, **2** Department of General Medicine,
Okayama University Graduate School of Medicine, Dentistry and Pharmaceutical Sciences, Okayama,
Japan, **3** Department of Pharmaceutical Biomedicine, Okayama University Graduate School of Medicine,
Dentistry and Pharmaceutical Sciences, Okayama, Japan, **4** Department of Transitional and Palliative Care,
Iizuka Hospital, Iizuka, Japan

* otsuka@s.okayama-u.ac.jp

**Data Availability Statement:** The minimal dataset necessary to replicate our study findings can be accessed directly through the Ministry of Health, Labour and Welfare's website (https://www.mhlw.go.jp/english/database/db-hw/outline/index.html).

## Abstract

### Background

In the global aging, the coronavirus disease 2019 (COVID-19) pandemic may have affected the place of death (PoD) in Japan, where hospital deaths have dominated for decades. We analyzed the PoD trends before and during the COVID-19 pandemic in Japan.

### Methods

This nationwide observational study used vital statistics based on death certificates from Japan between 1951 and 2021. The proportion of PoD; deaths at home, hospitals, and nursing homes; and annual percentage change (APC) were estimated using joinpoint regression analysis. Analyses were stratified by age groups and causes of death.

### Results

After 2019, home deaths exhibited upward trends, while hospital death turned into downward trends. By age, no significant trend change was seen in the 0–19 age group, while hospital deaths decreased in the 20–64 age group in 2019. The trend change in home death in the ≥65 age group significantly increased since 2019 with an APC of 12.3% (95% confidence interval [CI]: 9.0 to 15.7), while their hospital death trends decreased by −4.0% (95% CI: −4.9 to −3.1) in 2019−2021. By cause of death, home death due to cancer and the old age increased since 2019 with an APC of 29.3% (95% CI: 25.4 to 33.2) and 8.8% (95% CI: 5.5 to 12.2), respectively.

### Conclusion

PoD has shifted from hospital to home during the COVID-19 pandemic in Japan. The majority of whom were older population with cancer or old age.

This link leads to the comprehensive database which contains the data we have utilized.

**Funding:** The authors received no specific funding for this work.

**Competing interests:** The corresponding author has co-authored with the proposed editors for less than 5 years; however, this does not alter our adherence to the PLOS ONE policies on sharing data and materials.

## Introduction

With the increase in life expectancy, the global population is aging rapidly. Japan is one of the most super-aged countries in the world, and end-of-life care for older population has been a pressing issue [1]. Since the 1950s, medical services and technology have developed sharply during the period of rapid economic growth and the bubble economy in Japan. Consequently, we benefited from the increased number of inpatient beds, leading to a change in the place of death (PoD), *i.e.*, an increase in hospital deaths over the 21st century [2]. However, with the establishment of a long-term care insurance system in 2000 and the institutionalization of home care support clinics in 2006, home deaths have increased in Japan [3]. In fact, in 2005, we observed a decreasing trend in hospital deaths and an inverse increase in home deaths in Japan [4]. However, previous efforts have revealed that the decreasing trend of hospital deaths differs greatly depending on the cause of death (CoD) [5, 6], and the details of this trend need to be examined individually.

The coronavirus disease 2019 (COVID-19) pandemic occurred amid this trend in PoD. Facing the disease outbreak, the world has witnessed various changes in people's access to and utilization of healthcare services [7]. Although Japan recorded fewer COVID-19-related deaths than other countries [8], the provision of medical services for non-COVID-19 diseases, including emergency care, surgery, and hospitalization, is considerably limited [9–12]. Moreover, family visits to hospitalized patients were severely regulated from the perspective of infection prevention and control; thus, some people might refuse hospitalization. These direct and indirect changes could have contributed to the changing trend of PoD in Japan. In order to be prepared for the upcoming pandemic, the governments and health care providers need to know how the COVID-19 pandemic has changed the demographics of mortality, including PoD. However, studies on this topic are limited. Thus, this study aimed to determine the PoD and CoD trends before and during the COVID-19 pandemic in Japan, a country with a super-aged society.

## Materials and methods

### Data source

This population-based observational study was conducted in Japan using vital statistics. Data on the number of deaths by location and cause were obtained from vital statistics based on death certificates collected by the Japanese Ministry of Health, Labor, and Welfare from 1951 to 2021 [13]. In the Japanese death certificate database, the direct and underlying CoD and PoD are recorded based on information from death certificates completed by doctors within one week of death. The underlying CoD is published in the vital statistics, based on death certificates. Since 1995, it has been classified based on the International Classification of Diseases Tenth Revision (ICD-10) codes. Determining the underlying cause of death from death certificates consists of an auto-coding system, a rule-based process with manual review. Manual review is performed when the auto-coding system cannot assign an ICD-10 code or when ancillary information is included, which accounts for approximately 40% of the approximately 100,000 death certificates per month [14].

### Data processing

PoD was classified as hospital (hospital or physician's office), nursing home (care home or nursing care home), or home. Data were stratified by age as follows: 0–19 years, 20–64 years, and ≥65 years. The top five CoDs were redefined based on ICD-10 codes as follows: pneumonia (J12–J18), cerebrovascular disease (I60–I69), heart disease (I01–I02, I05–I09, I20–I25, I27,

and I30–I52), cancer (C00–97), and old age (R54). In 2021, the major causes of death were 26.5% from cancer, 14.9% from heart disease, 10.6% from old age, 7.3% from cerebrovascular disease, and 5.1% from pneumonia [13]. In the Japanese manual on completing a death certificate, old age is defined as the death of an old person from natural causes without an apparently describable CoD.

## Statistical analysis

Firstly, we analyzed the proportion of PoD from 1951 to 2021, then by age group or by the top five CoDs from 2001 to 2021. The percentage of PoDs was calculated by dividing the number of deaths that occurred at each hospital, nursing home, and home by the total number of deaths that occurred in one year. To estimate PoD trends, a joinpoint regression model was applied using the Joinpoint Regression Program version 4.9.1.0 (April 2022; National Cancer Institute). This study used the permutation test as the model selection method in the Joinpoint regression analysis. The year was the independent variable.

The annual percent change (APC) characterizes trends in the proportion of deaths by PoD over time. This method defines the percentage of deaths by PoD as a constant percentage change relative to the previous year's percentage. A single APC can accurately characterize the trend across a set of data. The Joinpoint model uses statistical criteria to determine when and how often an APC changes. Average Annual Percent Change (AAPC) summarizes trends over the entire study period. This allows the average AAPC over multiple years to be represented by a single number; the AAPC is useful when the joinpoint model shows a change in trend over those years. Additionally, it is calculated as a weighted average of the APCs from the joinpoint model, with the weight equal to the length of the APC interval. Data analysis was performed independently by M. Shibata and Y. Otsuka for validation. Statistical significance was set at $p < 0.05$.

## Ethics approval

The Japanese Ministry of Health, Labour, and Welfare and the Statistics Bureau of the Ministry of Internal Affairs and Communications provided the data for this study. The Okayama University Hospital Ethics Committee determined that a formal ethics review was unnecessary because the data were anonymous and accessible to the general public.

## Results

In 1951, hospital, home, and nursing home deaths accounted for 11.6%, 82.5%, and 0% of all deaths, respectively; in 2021, the percentages changed to 67.4%, 17.2%, and 13.5%, respectively. Trends in the number of PoDs over the past 70 years in Japan are presented in **Fig 1**. Since the 1950s, hospital deaths have steadily increased, whereas home deaths have gradually declined until the mid-2000s; however, they have gradually increased after that. The number of deaths in nursing homes has increased since the late 2000s. These trends indicated apparent shifts between 2019 and 2020: hospital deaths demonstrated a downward trend, whereas home deaths exhibited upward trends.

The number and the proportion of PoD were different by age in 2001 and 2021 (**S1 Table**). The PoD trend during this period was calculated by age group and represented in **Table 1**. The pediatric population aged ≤19 years demonstrated a monotonic trend, with an APC of 3.0% (95% CI: 2.6 to 3.5) increase in home deaths and 0.4% (95% CI: −0.7 to −0.2) decrease in hospital deaths. No trend change points were observed. In contrast, in young- and middle-aged adults (20–64 years), the proportion of home deaths remained slightly increasing trends by 2019 and indicated an upsurge thereafter with an APC of 12.2% (95% CI: 8.9 to 15.6).

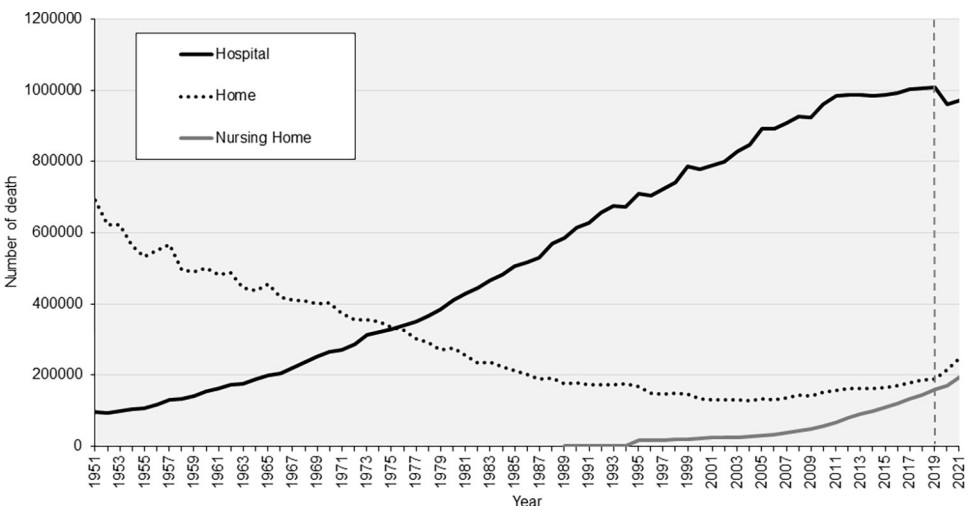

**Fig 1. Trends in the place of death since 1951 in Japan.** The number of counted deaths in hospitals, nursing homes, and homes are demonstrated. The number of hospital deaths decreased and that of other deaths increased after 2019 when the COVID-19 pandemic began.

Meanwhile, hospital deaths decreased suddenly from 2019 to 2021 with an APC of −5.0% (95% CI: −9.2 to −0.5). The older population aged ≥65 years initially demonstrated a decreasing trend in the early 2000s (APC, −4.1% 95% CI: −5.4 to −2.8), then rising trends from 2005 to 2019, and finally a big surge since 2019 with an APC of 12.3% (95% CI: 9.0 to 15.7). In the

**Table 1. Trends in the place of death by age group in Japan in 2001–2021.**

| Age/facility | Period 1 | | Period 2 | | Period 3 | | Period 4 | | Average APC (%) (95% CI) |
|---|---|---|---|---|---|---|---|---|---|
| | Years | APC (%) | Years | APC (%) | Years | APC (%) | Years | APC (%) | |
| **0–19 years** | | | | | | | | | |
| Home | 2001–2021 | 3.0* | | | | | | | 3.0* (2.6 to 3.5) |
| Hospital | 2001–2021 | −0.4* | | | | | | | −0.4* (−0.7 to −0.2) |
| Nursing home† | | – | | | | | | | – |
| **20–64 years** | | | | | | | | | |
| Home | 2001–2005 | 2.0* | 2005–2008 | 4.1* | 2008–2019 | 2.6* | 2019–2021 | 12.2* | 3.6* (3.0 to 4.2) |
| Hospital | 2001–2019 | −0.5* | 2019–2021 | −5.0* | | | | | −1.0* (−1.4 to −0.5) |
| Nursing home | 2001–2003 | −12.1 | 2003–2021 | 15.8* | | | | | 12.6* (9.3 to 16.1) |
| **≥65 years** | | | | | | | | | |
| Home | 2001–2005 | −4.1* | 2005–2015 | 0.3 | 2015–2019 | 2.2* | 2019–2021 | 12.3* | 0.9* (0.4 to 1.4) |
| Hospital | 2001–2005 | 0.5* | 2005–2009 | −0.6* | 2009–2019 | −1.1* | 2019–2021 | −4.0* | −1.0* (−1.1 to −0.9) |
| Nursing home | 2001–2003 | −0.9 | 2003–2007 | 6.7* | 2007–2013 | 12.7* | 2013–2021 | 7.6* | 8.0* (7.3 to 8.8) |

*Significantly different from zero ($p < 0.05$).

† Data were not available for this age group. APC, annual percentage change; CI, confidence interval.

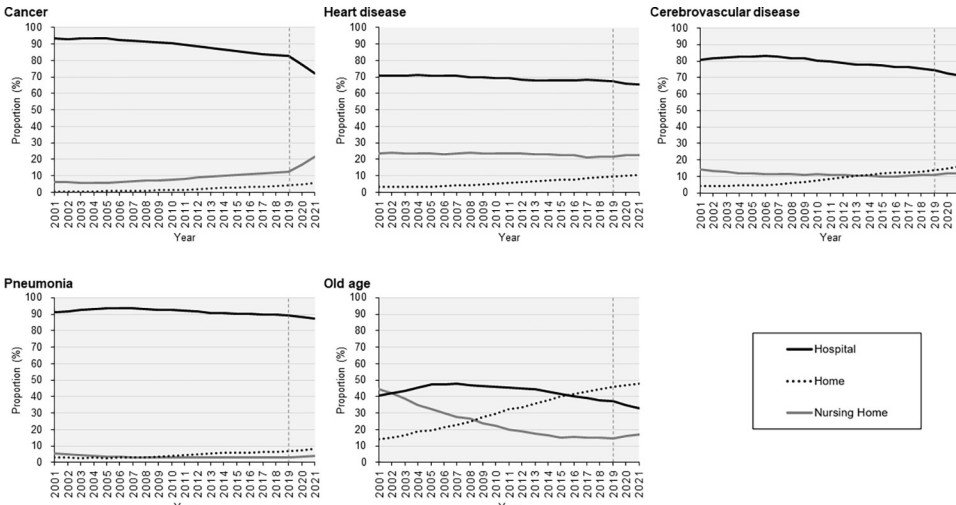

**Fig 2. Trends in the place of death by cause of death in Japan in 2001–2021.** The percentage of place of death among the top five CoD in Japan was determined. Statistically significant trend shifts in 2019 were observed for cancer and old age.

inverse proportion, hospital deaths in the older population decreased, resulting in an APC of −4.0% (95% CI: −4.9 to −3.1) in 2019–2021.

The trends in the proportions of PoD by CoD over the past two decades are depicted in **Fig 2**, and the results of the joinpoint regression analysis are summarized in **Table 2**. Regarding cancer-related deaths, the largest inflection point was observed in 2019. Proportion of home deaths had a significant increase since 2005, which further escalated between 2019 and 2021 with an APC of 29.3% (95% CI: 25.4 to 33.2). By contrast, that of hospital death started to decrease in 2005, and declined greatly in 2019 with an APC of −6.3% (95% CI: −6.9 to −5.8). No significant changes in heart disease were observed during the study period. The proportion of home deaths due to cerebrovascular disease initially demonstrated a decreasing trend and has increased since 2016 (APC, 3.4% 95% CI: 2.0 to 4.8). Hospital deaths followed an inverse trend and had an APC of −2.5% from 2019 to 2021. The proportion of pneumonia-related deaths indicated a trend similar to that of deaths related to cerebrovascular diseases. A remarkable increase in home deaths was observed from 2017 to 2021, with an APC of 5.9% (95% CI: 2.2 to 9.8), Finally, the proportion of home deaths due to old age initially followed significant decreasing trends, but indicated a great increase since 2019 (APC, 8.8% 95% CI: 5.5 to 12.2). Inversely proportional to this, the proportion of hospital deaths started to decline in 2005, with an APC of −0.8% (95% CI: −1.1 to −0.4) in 2005–2012, −2.8% (95% CI: −3.1 to −2.6) in 2012–2019, and −5.8% (95% CI: −7.1 to −4.5) in 2019–2021. The AAPC of all the categories for nursing home deaths demonstrated significant increases, ranging from 5.2% (95% CI: 4.5 to 5.9) for pneumonia to 12.4% (95% CI: 11.3 to 13.5) for cancer.

## Discussion

This study revealed a significant increase in home deaths after 2019 in Japan. This period coincides with the time of the COVID-19 pandemic. The first case of COVID-19 was identified in Japan on January 16, 2020, and the first emergency declaration was issued on April 7, 2020. Since then, the country has experienced waves 1 to 5 through 2021. The fifth wave (7/1/2021 to 9/30/2021), a delta strain (B.1.617), which tended to cause severe illness, was epidemic. This resulted in a sharp increase in the number of critical cases and caused the collapse of the medical system [15].

**Table 2. Trends in the place of death by cause of death in Japan in 2001–2021.**

| Age/facility | Period 1 | | Period 2 | | Period 3 | | Period 4 | | Average APC (%) (95% CI) |
|---|---|---|---|---|---|---|---|---|---|
| | Years | APC (%) | Years | APC (%) | Years | APC (%) | Years | APC (%) | |
| **Cancer** | | | | | | | | | |
| Home | 2001–2005 | −0.4 | 2005–2019 | 5.7* | 2019–2021 | 29.3* | | | 6.5* (6.0 to 7.1) |
| Hospital | 2001–2005 | 0.1 | 2005–2011 | −0.7* | 2011–2019 | −1.0* | 2019–2021 | −6.3* | −1.3* (−1.3 to −1.2) |
| Nursing home | 2001–2004 | 2.9 | 2004–2015 | 15.9* | 2015–2019 | 9.5* | 2019–2021 | 14.1* | 12.4* (11.3 to 13.5) |
| **Heart disease** | | | | | | | | | |
| Home | 2001–2013 | −0.1 | 2013–2017 | −2.1* | 2017–2021 | 0.9* | | | −0.3 (−0.3 to 0) |
| Hospital | 2001–2006 | 0 | 2006–2013 | −0.6* | 2013–2017 | 0.1 | 2017–2021 | −1.0* | −0.4* (−0.5 to −0.2) |
| Nursing home | 2001–2005 | 1.1 | 2005–2013 | 8.7* | 2013–2021 | 5.8* | | | 6.0* (5.4 to 6.6) |
| **Cerebrovascular disease** | | | | | | | | | |
| Home | 2001–2004 | −5.2* | 2004–2016 | −1.4* | 2016–2021 | 3.4* | | | −0.8* (−1.3 to −0.3) |
| Hospital | 2001–2006 | 0.5* | 2006–2019 | −0.8* | 2019–2021 | −2.5* | | | −0.7* (−0.8 to −0.5) |
| Nursing home | 2001–2006 | 2.5* | 2006–2013 | 12.2* | 2013–2019 | 4.5* | 2019–2021 | 8.4* | 7.0* (6.1 to 8.0) |
| **Pneumonia** | | | | | | | | | |
| Home | 2001–2006 | −10.2* | 2006–2017 | −0.6 | 2017–2021 | 5.9* | | | −1.9* (−2.7 to −1.0) |
| Hospital | 2001–2006 | 0.5* | 2006–2021 | −0.4* | | | | | −0.2* (−0.2 to −0.1) |
| Nursing home | 2001–2006 | −1.4 | 2006–2014 | 9.7* | 2014–2018 | 3.2* | 2018–2021 | 7.5* | 5.2* (4.5 to 5.9) |
| **Old age** | | | | | | | | | |
| Home | 2001–2014 | −7.6* | 2014–2019 | −2.0* | 2019–2021 | 8.8* | | | −4.7* (−5.1 to −4.3) |
| Hospital | 2001–2005 | 4.4* | 2005–2012 | −0.8* | 2012–2019 | −2.8* | 2019–2021 | −5.8* | −1.0* (−1.2 to −0.8) |
| Nursing home | 2001–2011 | 8.6* | 2011–2016 | 5.4* | 2016–2021 | 2.7* | | | 6.3* (6.0 to 6.6) |

*Significantly different from zero ($p < 0.05$). APC, annual percentage change; CI, confidence interval.

Based on the study result, the majority of the increase in home deaths was attributed to elderly individuals with cancer or old age, whereas COVID-19 did not constitute a large proportion of deaths at home. No change in the trend was observed in younger or older patients without malignancies. Thus, the shift in the PoD from hospital to home, recognized over the global outbreak of emerging infectious diseases, varied greatly by patient age and CoD. A multicenter web-based survey of the directors of home visit facilities in Japan has reported an increase in home visits for patients with cancer during the pandemic [16]. Additionally, a study using Japanese vital statistics data has reported an increase in excess deaths at home among patients with cancer since the early stages of the pandemic [17]. The present data reflect the results of previous studies.

Several reports on changes in the PoD during the COVID-19 pandemic have been published in the UK. A population-based modeling study has suggested a 220% increase in elderly

care facility deaths during the first 10 weeks of the pandemic [18]. Additionally, from March 2020 to March 2021, a descriptive analysis of mortality data from England, Wales, Scotland, and Northern Ireland has reported a 41% increase in home deaths and a 11% increase in hospital deaths, whereas hospice deaths decreased by 15% [19]. The UK has a significant number of COVID-19 deaths, with 130.1 deaths per 100,000 people [8]. Meanwhile, the mortality rate in Japan is 7.3 deaths per 100,000 people, with comparatively lower overall excess mortality [19]. Thus, the data between the two countries cannot be simply compared.

The changes in the PoD in Japan, where the direct impact of COVID-19 was minimal, may have been influenced by the following reasons. First, the pandemic severely limited hospital capacity. The pandemic's restrictions on access to hospital beds increased the number of cases in which hospitals could not accept emergency transports [20] and in cases where transport to hospitals providing intensive care was difficult [21]. This may have prevented patients who should have received care at the hospital from accessing it. Second, the intentions of the patient's family changed due to the restriction of visits. Although it is important to have family support in end-of-life care, hospital visitation restrictions were implemented to prevent hospital outbreaks during the pandemic. This may have increased the number of patients requesting home-based palliative care, and a multicenter web-based survey of home visit facilities in Japan during the pandemic supports this [16]. Furthermore, disease-specific differences in the prevalence of palliative care may explain why the impact of this shift varies by age and cause of death. Palliative care in Japan has a history of being developed mainly for patients with cancer under the Cancer Control Act [22] and home care and palliative care for patients without cancer is still developing. Therefore, it is possible that a system to provide home palliative care to the potential demand of patients without cancer, during the pandemic was not yet in place.

Strength of the present study is that it used the vital statistics and thus enabled us to clarify the nationwide trend of PoD in Japan. While, several limitations of this study should also be considered. First, this was a descriptive study, and the causal relationship between the COVID-19 pandemic and changes in the PoD is inconclusive. PoD may be affected by both environmental and individual variables, such as changes in patient preferences for care settings caused by the pandemic or the discharge of patients from medical institutions to secure hospital beds. However, the present findings cannot be directly used to assess causal correlations. Second, other unmeasured confounding factors that could potentially influence the choice of end-of-life care location in addition to age, sex, and CoD were not considered. Hence, other parameters identified in previous studies should be verified in future studies. However, PoD changes since the pandemic have occurred in only three years, and the impact of factors that occur over a longer period, such as population aging, is expected to be limited. Third, the CoDs on death certificates may have been misclassified. For example, in old age, any disease including COVID-19 present may be misclassified. Thus, the apparent increase in home deaths of old age may suggest an underdiagnosis of COVID-19. Nonetheless, the accuracy of CoD is rather high, particularly for patients with cancer whose changes in PoD were the most pronounced [23]. Finally, this study covered data from 2019 to 2021, including the first five waves of the COVID-19 pandemic in Japan. Depending on the viral variants and hospital capacity, the mortality rate of COVID-19 may change. Thus, the effects of the sixth and subsequent waves after 2022 are unknown. Future studies incorporating longer-term data are needed to determine the clinical significance of this trend change.

## Conclusions

In conclusion, the dying situation of Japanese patients with cancer or old age has changed during the COVID-19 pandemic. The rapid shift of PoD from hospital to home since the

pandemic may require greater expansion of home palliative care to accommodate the growing number of terminally ill homebound patients. Also, whether this trend is preferable for patients and their families and whether it will persist after the end of the pandemic remain debatable. Therefore, further studies focusing on the quality of end-of-life care during COVID-19 pandemic are suggested. At the very least, physicians are expected to continue to pursue a good death for patients.

## Supporting information

**S1 Table. Number and proportion of deaths by place and age group in 2001 and 2021.** (DOCX)

## Acknowledgments

We would like to thank Editage (www.editage.jp) for the English language editing.

## Author Contributions

**Conceptualization:** Masashi Shibata, Yuki Otsuka.

**Data curation:** Masashi Shibata, Yuki Otsuka.

**Formal analysis:** Masashi Shibata, Yuki Otsuka.

**Investigation:** Masashi Shibata, Yuki Otsuka.

**Methodology:** Hideharu Hagiya, Toshihiro Koyama.

**Supervision:** Yuki Otsuka, Hideharu Hagiya, Toshihiro Koyama, Hideyuki Kashiwagi, Fumio Otsuka.

**Writing – original draft:** Masashi Shibata, Yuki Otsuka, Hideharu Hagiya.

**Writing – review & editing:** Masashi Shibata, Yuki Otsuka, Hideharu Hagiya, Toshihiro Koyama, Hideyuki Kashiwagi, Fumio Otsuka.

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
