## [Decision Letter · Decision Letter 0]

23 Oct 2023

PONE-D-23-28375Changes in the place of death during COVID-19 pandemic in JapanPLOS ONE

Dear Dr. Otsuka,

Thank you for submitting your manuscript to PLOS ONE. After careful consideration, we feel that it has merit but does not fully meet PLOS ONE’s publication criteria as it currently stands. Therefore, we invite you to submit a revised version of the manuscript that addresses the points raised during the review process.

We look forward to receiving your revised manuscript.

Kind regards,

Mihajlo Jakovljevic, MD, PhD, MAE

Academic Editor

PLOS ONE

Journal Requirements:

"Yuki Otsuka has co-authored with the proposed editors for less than 5 years."

Reviewers' comments:

Reviewer's Responses to Questions

**Comments to the Author**

1. Is the manuscript technically sound, and do the data support the conclusions?

Reviewer #1: Yes

Reviewer #2: Partly

Reviewer #3: Yes

2. Has the statistical analysis been performed appropriately and rigorously? 

Reviewer #1: Yes

Reviewer #2: No

Reviewer #3: Yes

3. Have the authors made all data underlying the findings in their manuscript fully available?

Reviewer #1: Yes

Reviewer #2: Yes

Reviewer #3: No

4. Is the manuscript presented in an intelligible fashion and written in standard English?

Reviewer #1: Yes

Reviewer #2: Yes

Reviewer #3: Yes

5. Review Comments to the Author

Reviewer #1: The study addresses a very pertinent and timely issue, especially in the context of global aging and the challenges the COVID-19 pandemic has posed on health systems worldwide.

The findings are of significant importance as they illustrate the changing dynamics of the PoD in Japan, particularly during the pandemic years. The shift from hospital deaths to home deaths, especially in the older population with cancer or old age, is a trend that needs to be comprehensively understood for potential policy implications and healthcare planning. I find this manuscript to be of high quality and relevance. It sheds light on a topic that warrants immediate attention, especially in the current global health scenario.

Reviewer #2: I have reviewed the manuscript titled "Changes in the Place of Death During COVID-19 Pandemic in Japan." While the authors have made an attempt to explore the effects of the COVID-19 pandemic on the location of deaths in Japan, I have several major concerns regarding the methodology and the scientific significance of the study, which need to be addressed before considering publication.

The primary concern is the limited timeframe after COVID-19 pandemic in the study. The COVID-19 pandemic only spanned two years. To effectively evaluate the impact of COVID-19, it is desirable to have data covering both pre-pandemic (2019) and pandemic (2020) periods, with at least five years of data on each side [International Journal of Epidemiology, 2017, 348–355]. The short observation period may introduce bias, making it challenging to draw robust conclusions about the impact of COVID-19 on the changes observed. Furthermore, while the results may indicate APC, it is essential to consider whether these changes are clinically significant and meaningful absolute difference in the context of a 70-year timeframe.

The manuscript reports changes in the place of death, particularly an increase in home deaths due to cancer and old age. However, it lacks an in-depth analysis of the underlying reasons for these changes. A significant portion of the discussion relies on findings from other studies, which may not directly correlate with the data presented in this study. The discussion, therefore, appears to be speculative and lacking a scientific foundation.

The clinical implications of the study's findings remain unclear. It is challenging to ascertain what actionable insights or recommendations can be derived from this research, especially for future pandemic.

Reviewer #3: The paper analyzes the variation in the place of death (PoD) according to major causes of death during the COVID-19 pandemic in Japan using nationwide death certificate data. The results are interesting and contribute to the analysis of the effect of the pandemic on general and cause-specific mortality. The paper is well written and the results are clearly described. However, some clarifications in the methods section and some revisions of the results would improve the paper.

My comments and suggestions for the authors:

Materials and Methods

1) Line 73. Since the analysis concerns also cause-specific mortality data a more detailed description of death certificates data processing is required. In particular, specify whether death certificates are entirely processed through an automatic coding system (for the coding of the single entries and the selection of the underlying cause) or they are in part manually reviewed by experts (in this case, please report the percentage of certificates manually reviewed). In addition, since you are investigating long-time series data, you should specify whether the same coding system was adopted all over the study period, and whether there have been any change/update in the classification of causes of death (for example from ICD9 to ICD10). Lastly, it should specified that the analysis refers to underlying cause-of-death data (I suppose so).

2) Line 84, Statistical analysis. Please add details on the Jointpoint regression analysis, for instance minimum/maximum number of joinpoints tested, model used and regressor (calendar year). In addition, as some readers might not be familiar with this kind of analysis, I recommend to explain that the APC indicates the annual percent change in the proportion (?) of deaths by PoD and that it has been estimated for different time periods.

Results

3) Lines 100-101: “Trends in the number of PoDs over the past 70 years in Japan are presented in Fig. 1. Since the 1950s, hospital deaths have steadily increased, whereas home deaths have gradually declined”. From Fig.1 I would say that home deaths have gradually declined until mid 2000s, being gradually increasing thereafter.

4) lines 100-104: Please cite in the text the proportion of deaths by each single PoD on total deaths at the beginning and at the end of the study period. I would also suggest to include a table (in the main text or in the supplementary material) reporting the frequencies of deaths by PoD and their proportion in each age stratum, for the first year and the last year considered in the analysis i.e. 2001 and 2021 (see a proposal below), and to briefly comment on them before the presentation of the results of the joinpoint analysis.

2001 2021

no. of deaths % no. of deaths %

0-19 yrs

home

hospital

nursing home

20-64 yrs

home

hospital

nursing home

≥65 yrs

home

hospital

nursing home

5) Lines 111-112: “In order to detect the impact of COVID-19 more sensitively, the results of the trend analysis of the proportion of PoDs focused on the last 20 years in the 21st century”. This sentence expresses a methodological choice, therefore it should be moved to the methods section (Statistical analysis).

6) line 118: “ hospital deaths decreased suddenly in 2019…” it would be more appropriate to say “from 2019 to 2021” than “in 2019”

7) How much do the top five causes of death account on total mortality in Japan? This proportion should be explicitly mentioned in the text

Discussion

The change in time-trends of home deaths due to cancer or old age among the elderly population from 2019 is the main finding of the study.

8) Could the increase in home deaths due to old age be in part attributable to the possible under-diagnosis of SARS-CoV-2 infection, especially during the early pandemic phase? This aspect should be briefly discussed

9) I think the analysis of results could benefit from a brief description of the pandemic phases in Japan during 2020 and 2021

10) In the assessment of time-trends (in particular the increase in home/nursing home deaths) did you evaluate also the possible role of population aging? Although this effect is expected to be limited, this aspect should be discussed

6. PLOS authors have the option to publish the peer review history of their article (what does this mean?). If published, this will include your full peer review and any attached files.

Reviewer #1: No

Reviewer #2: **Yes: **Hiroyuki Ohbe

Reviewer #3: No

---

## [Author Response · Author response to Decision Letter 0]

15 Jan 2024

Reviewer #1

Comment: The study addresses a very pertinent and timely issue, especially in the context of global aging and the challenges the COVID-19 pandemic has posed on health systems worldwide.

The findings are of significant importance as they illustrate the changing dynamics of the PoD in Japan, particularly during the pandemic years. The shift from hospital deaths to home deaths, especially in the older population with cancer or old age, is a trend that needs to be comprehensively understood for potential policy implications and healthcare planning. I find this manuscript to be of high quality and relevance. It sheds light on a topic that warrants immediate attention, especially in the current global health scenario.

Response: We thank Reviewer #1 for the time and effort spent reviewing our manuscript and providing positive feedback. We are grateful for your assessment of our manuscript as high quality and relevant. 

Reviewer #2

Comment: I have reviewed the manuscript titled "Changes in the Place of Death During COVID-19 Pandemic in Japan." While the authors have made an attempt to explore the effects of the COVID-19 pandemic on the location of deaths in Japan, I have several major concerns regarding the methodology and the scientific significance of the study, which need to be addressed before considering publication.

Response: We thank Reviewer #2 for the time and effort spent reviewing our manuscript and for providing positive feedback and suggestions, which have helped us improve our manuscript considerably. We have answered each question below and hope that our responses and revisions address all your comments.

Comment: The primary concern is the limited timeframe after COVID-19 pandemic in the study. The COVID-19 pandemic only spanned two years. To effectively evaluate the impact of COVID-19, it is desirable to have data covering both pre-pandemic (2019) and pandemic (2020) periods, with at least five years of data on each side [International Journal of Epidemiology, 2017, 348–355]. The short observation period may introduce bias, making it challenging to draw robust conclusions about the impact of COVID-19 on the changes observed. Furthermore, while the results may indicate APC, it is essential to consider whether these changes are clinically significant and meaningful absolute difference in the context of a 70-year timeframe.

Response: Thank you for the comment. First, we applied Joinpoint analysis instead of ITS analysis. Your recommendation on covering “at least five years of data on each side” seems to be cited from the ITS analysis tutorial [International Journal of Epidemiology, 2017, 348–355], and we could not find a clear number of data points to be applied for that. The Joinpoint software we used takes the trend data and fits it to the simplest joinpoint model that allows all of its data points. In this case, we did not use 2019 as the split point but rather calculated the change points derived from the optimal model. However, this study only included the first five pandemic waves of COVID-19 in Japan, as you indicated. We have added a note in the Limitations section that future studies incorporating longer-term data are needed to determine clinical significance. (Lines 251‐252) In addition, as you pointed out, this study could not directly estimate the COVID-19 effects on PoD trends. We modified the title (Line 1), abstract (Lines 24 and 37), objectives (Lines 64-65) and discussion from this perspective (Lines 188-189, 256-258).

Comment: The manuscript reports changes in the place of death, particularly an increase in home deaths due to cancer and old age. However, it lacks an in-depth analysis of the underlying reasons for these changes. A significant portion of the discussion relies on findings from other studies, which may not directly correlate with the data presented in this study. The discussion, therefore, appears to be speculative and lacking a scientific foundation.

Response: Thank you for the suggestion. The paragraph discussing the potential reasons for the change in the PoD observed in this study has been revised. (Lines 216-231) 

Comment: The clinical implications of the study's findings remain unclear. It is challenging to ascertain what actionable insights or recommendations can be derived from this research, especially for future pandemic.

Response: We thank the reviewer for the insightful comments. The following information has been added to the conclusion: (Lines 256-258) 

Reviewer #3

Comment: The paper analyzes the variation in the place of death (PoD) according to major causes of death during the COVID-19 pandemic in Japan using nationwide death certificate data. The results are interesting and contribute to the analysis of the effect of the pandemic on general and cause-specific mortality. The paper is well-written, and the results are clearly described. However, some clarifications in the methods section and some revisions of the results would improve the paper.

Response: We thank Reviewer #3 for the time and effort review our manuscript and for providing positive feedback and suggestions, which have considerably helped us improve the manuscript. We have answered all the questions below and hope that our responses and revisions address all your comments.

Comment: Line 73. Since the analysis concerns also cause-specific mortality data a more detailed description of death certificates data processing is required. In particular, specify whether death certificates are entirely processed through an automatic coding system (for the coding of the single entries and the selection of the underlying cause) or they are in part manually reviewed by experts (in this case, please report the percentage of certificates manually reviewed). 

Response: Thank you for your insightful comments. In Japan, the process of determining the underlying cause of death from death certificates comprises an autocoding system, which is a rule-based process, and a manual review. A manual review is performed when an ICD-10 code cannot be assigned by the autocoding system or when ancillary information is included, which accounts for approximately 40% of 100,000 death certificates per month. This information was added to the Methods section. (Lines 76-80)

Comment: In addition, since you are investigating long-time series data, you should specify whether the same coding system was adopted all over the study period and whether there have been any change/update in the classification of causes of death (for example from ICD9 to ICD10). 

Response: Thank you for the comment. ICD10 has been used in Japanese Vital Statistics since 1995. Therefore, the same coding system was used in this study for PoD analysis based on the cause of death from 2001 to 2021. This information was added to the Methods section. (Lines 75-76)

Comment: Lastly, it should specified that the analysis refers to underlying cause-of-death data (I suppose so).

Response: Thank you for the comment. In Vital Statistics, the underlying CoD is published based on death certificates. This sentence has been added to the Methods section regarding the data sources. (Lines 74-75)

Comment: 2) Line 84 Statistical analysis. Please add details on the Jointpoint regression analysis, for instance minimum/maximum number of joinpoints tested, model used and regressor (calendar year). In addition, as some readers might not be familiar with this kind of analysis, I recommend to explain that the APC indicates the annual percent change in the proportion (?) of deaths by PoD and that it has been estimated for different time periods.

Response: Thank you for the comment. As you indicated, we have added detailed information on the selected model and variables, as well as the meaning of APC, to the Methods section. (Lines 99-110)

Comment: 3) Lines 100-101: “Trends in the number of PoDs over the past 70 years in Japan are presented in Fig. 1. Since the 1950s, hospital deaths have steadily increased, whereas home deaths have gradually declined”. From Fig.1 I would say that home deaths have gradually declined until mid 2000s, being gradually increasing thereafter.

Response: Thank you for the comment. As you have pointed out, this is correct. We have revised the sentence accordingly. (Lines 123-124)

Comment: 4) lines 100-104: Please cite in the text the proportion of deaths by each single PoD on total deaths at the beginning and at the end of the study period. I would also suggest to include a table (in the main text or in the supplementary material) reporting the frequencies of deaths by PoD and their proportion in each age stratum for the first year and the last year considered in the analysis i.e. 2001 and 2021 (see a proposal below), and to briefly comment on them before the presentation of the results of the joinpoint analysis.

2001 2021

no. of deaths % no. of deaths %

0-19 yrs

home

hospital

nursing home

20-64 yrs

home

hospital

nursing home

≥65 yrs

home

hospital

nursing home

Response: Thank you for the comment. We added the proportion of deaths by each PoD to the total deaths in 1951 and 2021. We have also created a new table as supplementary material, which reports the number and proportion of deaths by place and age group in 2001 and 2021, and briefly commented on them before presenting the results of the joinpoint analysis. (Table S1, Line 134)

Comment: 5) Lines 111-112: “In order to detect the impact of COVID-19 more sensitively, the results of the trend analysis of the proportion of PoDs focused on the last 20 years in the 21st century.””. This sentence expresses a methodological choice, therefore it should be moved to the methods section (Statistical analysis).

Response: Thank you for the comment. We moved this sentence to the Methods section and revised it accordingly. (Lines 94-95)

Comment: 6) line 118: “ hospital deaths decreased suddenly in 2019…” it would be more appropriate to say “from 2019 to 2021” than “in 2019”

Response: Thank you for the comment. We have revised the sentence accordingly. (Line 141)

Comment: 7) How many ofmuch do the top five causes of death account on total mortality in Japan? This proportion should be explicitly mentioned in the text.

Response: Thank you for the comment. We have shown that the proportion of the top five CoDs accounts for the total mortality in the Methods section. (Lines 87-89).

Comment: 8) Could the increase in home deaths due to old age be in part attributable to the possible underdiagnosis of SARS-CoV-2 infection, especially during the early pandemic phase? This aspect should be briefly discussed

Response: Thank you for the comment. We have mentioned the possibility of misclassifying the cause of death in the Limitations section and added your insightful perspective. (Lines 244-246)

Comment: 9) I think the analysis of results could benefit from a brief description of the pandemic phases in Japan during 2020 and 2021.

Response: Thank you for the comment. We added a brief description of the phases of the COVID-19 pandemic in Japan in 2020 and 2021 to the Discussion section. (Lines 189-194)

Comment: 10) In the assessment of time trends (in particular, the increase in home/nursing home deaths), did you evaluate also the possible role of population aging? Although this effect is expected to be limited, this aspect should be discussed

Response: Thank you for the comment. We have added your insightful perspective to the Limitations section. (Lines 242-243)

---

## [Decision Letter · Decision Letter 1]

15 Feb 2024

Changes in the place of death before and during the COVID-19 pandemic in Japan

PONE-D-23-28375R1

Dear Dr. Otsuka,

We’re pleased to inform you that your manuscript has been judged scientifically suitable for publication and will be formally accepted for publication once it meets all outstanding technical requirements.

Kind regards,

Mihajlo Jakovljevic, MD, PhD, MAE

Academic Editor

PLOS ONE

Additional Editor Comments (optional):

Reviewers' comments:

Reviewer's Responses to Questions

**Comments to the Author**

1. If the authors have adequately addressed your comments raised in a previous round of review and you feel that this manuscript is now acceptable for publication, you may indicate that here to bypass the “Comments to the Author” section, enter your conflict of interest statement in the “Confidential to Editor” section, and submit your "Accept" recommendation.

Reviewer #3: All comments have been addressed

2. Is the manuscript technically sound, and do the data support the conclusions?

Reviewer #3: Yes

3. Has the statistical analysis been performed appropriately and rigorously? 

Reviewer #3: Yes

4. Have the authors made all data underlying the findings in their manuscript fully available?

Reviewer #3: No

5. Is the manuscript presented in an intelligible fashion and written in standard English?

Reviewer #3: Yes

6. Review Comments to the Author

Reviewer #3: The authors have properly addressed the comments and they have revised the manuscript accordingly.

The paper has improved and I suggest to accept it for publication

7. PLOS authors have the option to publish the peer review history of their article (what does this mean?). If published, this will include your full peer review and any attached files.

Reviewer #3: No

---

## [Editor Report · Acceptance letter]

19 Feb 2024

PONE-D-23-28375R1 

PLOS ONE

Dear Dr. Otsuka, 

I'm pleased to inform you that your manuscript has been deemed suitable for publication in PLOS ONE. Congratulations! Your manuscript is now being handed over to our production team.

Kind regards, 

on behalf of

Professor Mihajlo Jakovljevic 

Academic Editor

PLOS ONE